# Classical Modelling of a Bosonic Sampler with Photon Collisions

**DOI:** 10.3390/e25020210

**Published:** 2023-01-21

**Authors:** Mikhail Umanskii, Alexey N. Rubtsov

**Affiliations:** Russian Quantum Center, Skolkovo, Moscow 143025, Russia

**Keywords:** boson sampling, photon collisions, computational complexity

## Abstract

The original formulation of the boson sampling problem assumed that little or no photon collisions occur. However, modern experimental realizations rely on setups where collisions are quite common, i.e., the number of photons *M* injected into the circuit is close to the number of detectors *N*. Here we present a classical algorithm that simulates a bosonic sampler: it calculates the probability of a given photon distribution at the interferometer outputs for a given distribution at the inputs. This algorithm is most effective in cases with multiple photon collisions, and in those cases, it outperforms known algorithms.

## 1. Introduction

Quantum computers are computational devices that operate using phenomena described by quantum mechanics. Therefore, they can carry out operations that are not available for classical computers. Practical tasks are known which can be solved exponentially faster using quantum computers rather than classical ones. For example, the problem of integer factorization, which underlies the widely used RSA cryptosystem, can be solved by classical computers only in an exponential number of operations, whereas the quantum Shor’s algorithm [1] can solve it in a polynomial number of operations. Due to the technological challenges of manufacturing quantum computers, quantum supremacy (the ability of a quantum computational device to solve problems that are intractable for classical computers) in practice remains an open question.

Boson sampling [2] is a good candidate for demonstrating quantum supremacy. Basically, boson samplers are linear-optical devices containing a number of non-classical sources of indistinguishable photons, a multichannel interferometer mixing photons of different sources, and photon detectors at the output channels of the interferometer. A more specific set-up to be addressed in this paper deals with the single photon sources. In this case, there are exactly *M* single photons injected into some or all *N* inputs of the interferometer. Performing multiple measurements of the photon counts at the outputs, one characterizes experimentally the many-body quantum statistics after the interferometer [3], given an input state and the interferometer matrix.

Boson samplers are not universal quantum computers, that is they cannot perform arbitrary unitary rotations in a high-dimensional Hilbert space of a quantum system. Nevertheless, a simulation of a boson sampler with a classical computer requires a number of operations exponential in *M*. It was shown [4] that a classical complexity of the boson sampling matches the complexity of computing the permanent of a complex matrix. This means that the problem of a boson sampling is #P-hard [5] and there are no known classical algorithms that solve it in polynomial time. The best known exact algorithm for computing the permanent of a n×n matrix is the Ryser formula [6], which requires $O(n2^{n})$ operations. The Clifford–Clifford algorithm [7] is known to solve the boson sampling problem in $O(M2^{M}+NM^{2})$ operations. This makes large enough bosonic samples practically intractable with classical computational devices. Although boson sampling does not allow for arbitrary quantum computations, there are still practical problems that can be solved with boson sampling: for example, molecular docking [8], calculating the vibronic spectrum of a molecule [9,10] as well as certain problems in graph theory [11]. Boson sampling is also useful for statistical modeling [12] and machine learning [13,14].

There are several variants of boson sampling that aim at improving the photon generation efficiency and increasing the scale of implementations. For example, Scattershot boson sampling uses many parametric down-conversion sources to improve the single photon generation rate. It has been implemented experimentally using a 13-mode integrated photonic chip and six PDC photon sources [15]. Another variant is the Gaussian boson sampling [2,16], which uses the Gaussian input states instead of single photons. The Gaussian input states are generated using PDC sources, and it allows a deterministic preparation of the non-classical input light sources. In this variant, the relative input photon phases can affect the sampling distribution. Experiments were carried out with N=12 [17] and N=100 [18]. The latter implementation uses the PPKTP crystals as PDC sources and employs an active phase-locking mechanism to ensure a coherent superposition.

Any experimental set-up, of course, differs from the idealized model considered in theoretical modeling. Bosonic samplers suffer from two fundamental types of imperfections. First, the parameters of a real device, such as the reflection coefficients of the beam splitters and the phase rotations, are never known exactly. Varying the interferometer parameters too much can change the sampling statistics drastically so that modeling of an ideal device no longer makes much sense. Assume now that we know the parameters of the experimental set-up with great accuracy. Then what makes the device non-ideal is primarily photon losses, that is, not all photons emitted at the inputs are detected in the output channels. These losses happen because of imperfections in photon preparation, absorption inside the interferometer and imperfect detectors. There are different ways of modeling losses, for example by introducing the extra beam splitters [19] or replacing the interferometer matrix by a combination of lossless linear optics transformations and the diagonal matrix that contains transmission coefficients that are less than one [20].

Imperfections in middle-sized systems make them, in general, easier to emulate with classical computers [21]. It was shown [22] that with the increase of losses in a system the complexity of the task decreases. When the number of photons $M^{\prime }$ that arrive at the outputs is less than $\sqrt{M}$, the problem of a boson sampling can be efficiently solved using classical computers. On the other hand, if the losses are low, the problem remains hard for classical computers [23].

Photon collisions are a particular phenomenon that is present in nearly any experimental realization but were disregarded in the proposal by Aaronson and Arkhipov [4]. Originally it was proposed that the number of the interferometer channels is roughly a square of the number of photons in the set-up, $N\geq M^{2}$. In this situation, all or most of the photons arrive at a separate channel, that is no or a few of the photon collisions occur. In the experimental realizations [18], however, N≈M.

Generally, a large number of photon collisions makes the system easier to emulate. For example, one can consider the extreme case that all photons arrive at a single output channel – the probability of such an outcome can be estimated within a polynomial time. The effect of the photon collisions on the computational complexity of a boson sampling has been previously studied [24]. A measure called the Fock state concurrence sum was introduced and it was shown that the minimal algorithm runtime depends on this measure. There is an algorithm for the Gaussian boson sampling that also takes advantage of photon collisions [25].

In this paper, we present the algorithm aimed at the simulation of the bosonic samplers with photon collisions. In the N≈M regime, our scheme outperforms the Clifford–Clifford method. For example, we consider an output state of the sampler with M=N where one-half of the outputs are empty, and the other half is populated with 2 photons in each channel. Computing the probability of such an outcome requires us $O\left(N^{2}3^{N/2}\right)$ operations. The speedup in states that have more collisions is even greater.

## 2. Problem Specification

Consider a linear-optics interferometer with *N* inputs and *N* outputs which is described by a given unitary N×N matrix *U*:(1)$$b_{i}^{†}=\sum_{j=1}^{N}u_{ij}a_{j}^{†},\;a_{i}^{†}=\sum_{j=1}^{N}u_{ji}^{*}b_{j}^{†},$$
where $a_{i}^{†}$ and $b_{i}^{†}$ are the creation operators acting on inputs and outputs, respectively. We will denote the input state as
(2)$$\left|k\rangle =\right|k_{1},k_{2},\ldots ,k_{N}\rangle =\prod_{i=1}^{N}\frac{1}{\sqrt{k_{i}!}}{\left(a_{i}^{†}\right)}^{k_{i}}|0,0,\ldots ,0\rangle ,$$
where $k_{i}$ is the number of photons in the *i*-th input. The output state will be denoted as
$$\left|l\rangle =\right|l_{1},l_{2},\ldots ,l_{N}\rangle =\prod_{i=1}^{N}\frac{1}{\sqrt{l_{i}!}}{\left(b_{i}^{†}\right)}^{l_{i}}|0,0,\ldots ,0\rangle .$$

It follows from (1) and (2) that a specific input state corresponds to a set of output states that are observed with different probabilities:$$|k_{1},k_{2},\ldots ,k_{N}\rangle =\prod_{i=1}^{N}\frac{1}{\sqrt{k_{i}!}}{\left(a_{i}^{†}\right)}^{k_{i}}|0,0,\ldots ,0\rangle =\prod_{i=1}^{N}\frac{1}{\sqrt{k_{i}!}}{\left(\sum_{j=1}^{N}u_{ji}^{*}b_{j}^{†}\right)}^{k_{i}}|0,0,\ldots ,0\rangle$$

The product $\prod_{i=1}^{N}{\left(\sum_{j=1}^{N}u_{ji}^{*}b_{j}^{†}\right)}^{k_{i}}$ can be written as
(3)$$\prod_{i=1}^{N}{\left(\sum_{j=1}^{N}u_{ji}^{*}b_{j}^{†}\right)}^{k_{i}}={\left(u_{11}^{*}b_{1}^{†}+u_{21}^{*}b_{2}^{†}+\ldots +u_{N1}^{*}b_{N}^{†}\right)}^{k_{1}}\cdot \ldots \cdot {\left(u_{1N}^{*}b_{1}^{†}+u_{2N}^{*}b_{2}^{†}+\ldots +u_{NN}^{*}b_{N}^{†}\right)}^{k_{N}}.$$

After expanding, this expression will be a sum of terms that have the following form:$$\alpha \left(l_{1},l_{2},\ldots ,l_{N}\right)\prod_{i=1}^{N}\frac{1}{\sqrt{k_{i}!}}{\left(b_{i}^{†}\right)}^{l_{i}}|0,0,\ldots ,0\rangle =\alpha \left(l_{1},l_{2},\ldots ,l_{N}\right)\prod_{i=1}^{N}\sqrt{\frac{l_{i}!}{k_{i}!}}|l_{1},l_{2},\ldots ,l_{N}\rangle ,$$
where $\alpha (l_{1},l_{2},\ldots ,l_{N})$ is a complex number that consists of the elements of *U* that correspond to the given output state. Therefore, the probability of observing an output state $|l_{1},l_{2},\ldots ,l_{N}\rangle$ will be
$${\left|\langle l_{1},l_{2},\ldots ,l_{N}k_{1},k_{2},\ldots ,k_{N}\rangle \right|}^{2}={\left|\alpha \left(l_{1},l_{2},\ldots ,l_{N}\right)\prod_{i=1}^{N}\sqrt{\frac{l_{i}!}{k_{i}!}}\right|}^{2}={\left|\alpha (l_{1},l_{2},\ldots ,l_{N})\right|}^{2}\prod_{i=1}^{N}\frac{l_{i}!}{k_{i}!}.$$

The problem is to determine the probabilities of all of the output states. The main difficulty lies in calculating the number $\alpha (l_{1},l_{2},\ldots ,l_{N})$ for the given input and output states. This paper presents an algorithm that solves this problem using the properties of the Fourier transform.

## 3. Algorithm Description

Let us define a function
(4)$$g\left(t;\left\{Q_{i}\right\}\right)=\prod_{p=1}^{N}{\left(\sum_{q=1}^{N}e^{i2\pi Q_{q}t}\;u_{qp}^{*}\right)}^{k_{p}},$$
where $\left\{Q_{i}\right\}$ is some fixed set of *N* natural numbers. The choice of $\left\{Q_{i}\right\}$ will later be discussed in detail. This function represents the expression (3), where the creation operators $b_{j}^{†}$ are replaced with exponents $e^{i2\pi Q_{j}t}$ that oscillate with frequencies $Q_{j}$.

After expanding the expression (4), we get
$$g\left(t;\left\{Q_{i}\right\}\right)=\prod_{p=1}^{N}{\left(\sum_{q=1}^{N}e^{i2\pi Q_{q}t}\;u_{qp}^{*}\right)}^{k_{p}}$$
$$={\left(e^{i2\pi Q_{1}t}u_{11}^{*}+e^{i2\pi Q_{2}t}u_{21}^{*}+\ldots +e^{i2\pi Q_{N}t}u_{N1}^{*}\right)}^{k_{1}}\cdot \ldots \cdot {\left(e^{i2\pi Q_{1}t}u_{1N}^{*}+e^{i2\pi Q_{2}t}u_{2N}^{*}+\ldots +e^{i2\pi Q_{N}t}u_{NN}^{*}\right)}^{k_{N}}$$
$$=\sum e^{i2\pi \sum_{i=1}^{N}l_{i}Q_{i}t}\;\alpha \left(l_{1},l_{2},\ldots ,l_{N}\right),$$
where the sum is computed over all sets $\left\{l_{1},\ldots ,l_{N}\right\}$ such that $\sum_{i=1}^{N}l_{i}=M$.

Therefore, for each possible output state $\left|l\rangle =\right|l_{1},l_{2},\ldots ,l_{N}\rangle$ there is the harmonic in $g(t;\left\{Q_{i}\right\})$ that has a frequency of $f(|l\rangle ;\left\{Q_{i}\right\})=\sum_{i=1}^{N}l_{i}Q_{i}$ and an amplitude of $\alpha (l_{1},l_{2},\ldots ,l_{N})$. The set of numbers $\left\{Q_{i}\right\}$ can be chosen in such a way that the harmonics do not overlap, i.e., there are no two outputs states |l〉 and $|l^{\prime }\rangle$ with equal frequencies $f(|l\rangle ;\left\{Q_{i}\right\})=f(|l^{\prime }\rangle ;\left\{Q_{i}\right\})$.

If no harmonics overlap, then any of the numbers $\alpha (l_{1},l_{2},\ldots ,l_{N})$ can be found from the Fourier transform of the function $g(t;\left\{Q_{i}\right\})$. On the other hand, to calculate the probability of a specific state |l〉 it is sufficient to choose $\left\{Q_{i}\right\}$ in such a way that the frequency $f(|l\rangle ;\left\{Q_{i}\right\})$ is unique in the spectrum, i.e., the frequency of any other state $|l^{\prime }\rangle$ differs from the frequency of the state in consideration: $f(|l\rangle ;\left\{Q_{i}\right\})\neq f(|l^{\prime }\rangle ;\left\{Q_{i}\right\})\;\forall |l^{\prime }\rangle \neq |l\rangle$.

An example of $g(t,\left\{Q_{i}\right\})$ with non-overlapping harmonics and its spectrum can be seen in Figure 1 (the choice of $\left\{Q_{i}\right\}$ used here is described in Section 3.1).

### 3.1. The First Method of Choosing $\left\{Q_{i}\right\}$

Let us consider the methods of choosing $\left\{Q_{i}\right\}$ that will satisfy the necessary conditions on the spectrum. The first one consists of the following: let *M* be the total number of photons at the inputs, i.e., $M=\sum_{i=1}^{N}k_{i}$ for the input state $|k_{1},k_{2},\ldots ,k_{N}\rangle$. We choose $Q=\{1,M+1,{\left(M+1\right)}^{2},\ldots ,{\left(M+1\right)}^{N-1}\}$, or $Q_{i}={\left(M+1\right)}^{i-1}$. Then for an any output state $|l_{1},l_{2},\ldots ,l_{N}\rangle$ the sum
$$\sum_{i=1}^{N}l_{i}Q_{i}=1\cdot l_{1}+\left(M+1\right)\cdot l_{2}+{\left(M+1\right)}^{2}\cdot l_{3}+\ldots +{\left(M+1\right)}^{N-1}\cdot l_{N}$$
will be a number that has a representation $\bar{l_{N}l_{N-1}\ldots l_{1}}$ in a positional numeral system with a radix M+1 (since $l_{i}<M+1\;\forall i$). From the uniqueness of representation of numbers in positional numeral systems, it follows that every sum $\sum_{i=1}^{N}l_{i}Q_{i}$ (some number in a positional numeral system with a radix M+1) will correspond to exactly one set of numbers $l_{1},l_{2},\ldots ,l_{N}$ (its representation in this numeral system; $l_{i}$ being its digits).

Using this method of choosing $\left\{Q_{i}\right\}$ guarantees that the probability of any output state can be calculated from the spectrum of $g(t;\left\{Q_{i}\right\})$, since the frequencies $f(|l\rangle ;\left\{Q_{i}\right\})$, $f(|l^{\prime }\rangle ;\left\{Q_{i}\right\})$ are different for any two output states $\left|l\rangle \neq \right|l^{\prime }\rangle$.

### 3.2. The Second Method of Choosing $\left\{Q_{i}\right\}$

Another method of choosing $\left\{Q_{i}\right\}$ is useful when the goal is to compute the probability of one specific output state |l〉 when the input state |k〉 is given. This method does not guarantee that the frequencies will be different for any two output states, but it guarantees that the frequency of the state in consideration (the target frequency) $f(|l\rangle ;\left\{Q_{i}\right\})$ will be unique in the spectrum. Note that in this case $Q_{i}=Q_{i}(|l\rangle )$, i.e., the choice of $\left\{Q_{i}\right\}$ depends on the output state.

This method of choosing $\left\{Q_{i}\right\}$ can be described in the following way:(5)$$Q_{i}=\left\{\begin{array}{c} \prod_{j=1}^{i-1}\left(l_{j}+1\right),\;l_{i}\neq 0\\ 0,\;l_{i}=0; \end{array}\right.$$
$$Q_{1}=\left\{\begin{array}{c} 1,\;l_{1}\neq 0\\ 0,\;l_{1}=0. \end{array}\right.$$

Therefore if all of the outputs in the state |l〉 contain the photons, then $Q_{1}=1$, $Q_{2}=\left(l_{1}+1\right)$, $Q_{3}=\left(l_{2}+1\right)\left(l_{1}+1\right)$ and so on: $Q_{i+1}$ is $\left(l_{i}+1\right)$ times greater than $Q_{i}$.

Let us show that this method will actually lead to the target frequency being unique in the spectrum. Let $\left\{Q_{i}\right\}=\left\{Q_{i}(|l\rangle \right)\}$ be the frequencies calculated using the method described above. We need to prove that for any output state $|l^{\prime }\rangle \neq |l\rangle$ it is true that $f(|l\rangle ;\left\{Q_{i}\right\})\neq f(|l^{\prime }\rangle ;\left\{Q_{i}\right\})$, i.e.,
$$\sum_{i=1}^{N}l_{i}Q_{i}\neq \sum_{i=1}^{N}l_{i}^{\prime }Q_{i}.$$

Firstly, let us suppose that some of the outputs in the state $|l_{1},l_{2},\ldots ,l_{N}\rangle$ contain 0 photons. Let $h_{1},\ldots ,h_{K}$ be the indices of the outputs that contain at least one photon: $l_{h_{i}}>0\;\forall i\in \left\{1,2,\ldots ,K\right\};\;\;K<N$. Then the condition $f(|l\rangle ;\left\{Q_{i}\right\})\neq f(|l^{\prime }\rangle ;\left\{Q_{i}\right\})$ becomes
$$\sum_{i=1}^{K}l_{h_{i}}Q_{h_{i}}\neq \sum_{i=1}^{K}l_{h_{i}}^{\prime }Q_{h_{i}},$$
since all the terms corresponding to empty outputs are zero in both sums ($Q_{i}=0$ if the *i*-th output contains 0 photons).

Note that we can view it as a “reduced” system with the *K* outputs, in which the output state |l〉 contains at least one photon in every output. However, this system has one difference. Previously we considered the possible output states to be all the states that satisfy $\sum_{i=1}^{N}l_{i}^{\prime }=M$ and $l_{i}^{\prime }\geq 0\;\forall i$. Now, in this “reduced” system we must consider all the output states such that $\sum_{i=1}^{K}l_{i}^{\prime }\leq M$ and $l_{i}^{\prime }\geq 0\;\forall i$. This happens because the output states $|l^{\prime }\rangle$ can have a non-zero amount of the photons in the outputs that were empty in |l〉; such outputs will have no effect on the frequency and they remain outside the “reduced” system.

Therefore, instead of a system where some outputs can be empty and some $Q_{i}$ can be zero, but $\sum_{i=1}^{N}l_{i}=\sum_{i=1}^{N}l_{i}^{\prime }$, we can consider the system where $l_{i}>0\;\forall i$ but $\sum_{i=1}^{N}l_{i}\geq \sum_{i=1}^{N}l_{i}^{\prime }$. In this system $Q_{i}=\prod_{j=1}^{i-1}\left(l_{j}+1\right)$, and $Q_{1}=1$.

To prove the correctness of the algorithm, we must prove the following statement:

**Theorem** **1.**
*Let N be some natural number. Let $l_{1},l_{2},\ldots ,l_{N},l_{1}^{\prime },l_{2}^{\prime },\ldots l_{N}^{\prime }$ be natural numbers that satisfy the following conditions:*

*(1) li>0,li′≥0∀i;*

*(2) ∑i=1Nli≥∑i=1Nli′;*

*(3) $\sum_{i=1}^{N}l_{i}Q_{i}=\sum_{i=1}^{N}l_{i}^{\prime }Q_{i}$, where $Q_{i}=\prod_{j=1}^{i-1}\left(l_{j}+1\right)$ (and $Q_{1}=1$).*

*Then li=li′∀i.*


The proof of this statement can be found in Appendix A.

## 4. Parameters of the Fourier Transform

To calculate the Fourier transform of the function $g(t;\left\{Q_{i}\right\})$ we will use a fast Fourier transform (FFT). Firstly, we will define its parameters: the sampling interval Δt (or the sampling frequency $f_{s}=\frac{1}{\Delta t}$) and the number of data points *K*. The function will be calculated at points nΔt,1≤n≤K. Since all the frequencies in the spectrum of $g(t;\left\{Q_{i}\right\})$ are natural numbers, they can be discerned with the frequency resolution of Δf=1. The function therefore will be calculated in the points within an interval [0;1] which contains at least one period of each harmonic.

The sampling frequency $f_{s}=\frac{1}{\Delta t}$ is often chosen according to the Nyquist-Shannon theorem: if the Nyquist frequency $f_{N}=\frac{f_{s}}{2}=\frac{1}{2\Delta t}$ is greater than the highest frequency in the spectrum $f_{max}$, then the function can be reconstructed from the spectrum and no aliasing occurs. Therefore, one way of choosing the sampling frequency is $f_{s}=2f_{max}$. It can be used with both methods of choosing $\left\{Q_{i}\right\}$.

Since the function is calculated in the points within the interval [0;1], the number of data points *K* is equal to the sampling frequency $f_{s}$. Optimization of the algorithm requires lowering the sampling frequency as much as possible.

If the goal is to calculate the probability of one specific state |l〉 and the second method of choosing $\left\{Q_{i}\right\}$ is used, then the sampling frequency $f_{s}$ can be chosen to be lower than $2f_{max}$. This will lead to aliasing: a peak with frequency *f* will be aliased by the peaks with frequencies $f+kf_{s},\;k\in Z$. To correctly calculate the probability of the output state from the spectrum computed this way, the spectrum must not contain frequencies that satisfy $f(|l^{\prime }\rangle ;\left\{Q_{i}\right\})=f(|l\rangle ;\left\{Q_{i}\right\})+kf_{s},\;k\in Z$. Note that it will not be possible to reconstruct the function $g(t;\left\{Q_{i}\right\})$ from such a spectrum.

We will show that the sampling frequency $f_{s}$ for calculating the probability of the output state |l〉 using the second method of choosing $\left\{Q_{i}\right\}$ can be chosen to be greater by one than the target frequency $f=f(|l\rangle ;\{Q_{i}(|l\rangle )\})$:$$f_{s}=f+1=\sum_{i=1}^{N}l_{i}Q_{i}+1$$

To prove this statement, we must show that the spectrum of $g(t;\left\{Q_{i}\right\})$ will not contain the frequencies $f^{\prime }=f(|l^{\prime }\rangle ;\left\{Q_{i}\right\})$ that satisfy $f^{\prime }=f+kf_{s},\;k\in Z$. This is shown by a theorem that is analogous to Theorem 1 yet has a weaker condition: equation in condition (3) is taken modulo $f_{s}$.

**Theorem** **2.**
*Let N be some natural number. Let $l_{1},l_{2},\ldots ,l_{N},l_{1}^{\prime },l_{2}^{\prime },\ldots l_{N}^{\prime }$ be natural numbers that satisfy the following conditions:*

*(1) li>0,li′≥0∀i;*

*(2) ∑i=1Nli≥∑i=1Nli′;*

*(3) $\sum_{i=1}^{N}l_{i}Q_{i}\equiv \sum_{i=1}^{N}l_{i}^{\prime }Q_{i}\;\mathrm{mod}\;\left(\sum_{i=1}^{N}l_{i}Q_{i}+1\right)$, where $Q_{i}=\prod_{j=1}^{i-1}\left(l_{j}+1\right)$ (and $Q_{1}=1$).*

*Then li=li′∀i.*


The proof of this statement can be found in Appendix A.

## 5. Complexity of the Algorithm

Let us consider the computational complexity of this algorithm. The complexity of a fast Fourier transform of a data array of *K* points is O(KlogK). The total complexity of the algorithm consists of the complexity of calculating $g(t;\left\{Q_{i}\right\})$ in *K* points and the complexity of a fast Fourier transform.

Computing $g(t;\left\{Q_{i}\right\})$ in each point is done in $\propto N^{2}$ operations: the expression for $g(t;\left\{Q_{i}\right\})$ consists of at most *N* factors, each of which can be computed in *N* additions, *N* multiplications and *N* exponentiations. If some of the inputs are empty, there will be fewer factors in the expression, and the resulting complexity will be lower.

When the first method of choosing $\left\{Q_{i}\right\}$ is used, the number of data points *K* is proportional to the highest frequency in the spectrum of $g(t;\left\{Q_{i}\right\})$, since the sampling frequency is chosen using the Nyquist-Shannon theorem. The frequencies corresponding to the outputs states, in this case, are equal to $\sum_{i=1}^{N}l_{i}Q_{i}=\sum_{i=1}^{N}l_{i}{\left(M+1\right)}^{i-1}$. The highest frequency then is $M{\left(M+1\right)}^{N-1}$ and corresponds to the state where the last output contains all the photons. The total complexity of calculating all the probabilities will then be
$$O\left(N^{2}M{\left(M+1\right)}^{N-1}\right)+O\left(M{\left(M+1\right)}^{N-1}\log\left(M{\left(M+1\right)}^{N-1}\right)\right)$$
$$=O\left(N^{2}M^{N}+NM^{N}\log M\right).$$

When the second method of choosing $\left\{Q_{i}\right\}$ is used, the number of data points *K* depends on the frequency of the output state in consideration. This frequency is highest when photons are spread over outputs evenly. For the system with M=mN, m∈N this corresponds to the state where each output contains *m* photons. In this case, the highest frequency is equal to
$$\sum_{i=1}^{N}m{\left(m+1\right)}^{i-1}=m\frac{{\left(m+1\right)}^{N-1}-1}{m}={\left(m+1\right)}^{N-1}-1.$$

Therefore, the sampling frequency and the required number of data points will be ${\left(m+1\right)}^{N-1}$. The complexity of the algorithm in the worst case will be
$$O\left(N^{2}{\left(m+1\right)}^{N}+N{\left(m+1\right)}^{N}\log\left(m+1\right)\right).$$

In most states, however, photons will not be spread evenly between outputs, and outputs with a high number of photons will lower the sampling frequency and the complexity for calculating the probability of the state. This means that the more photon collisions are in a state, the better this algorithm performs. Let us consider several specific cases.

1. M=N, the goal is to compute the output state that contains 2 photons in one half of the outputs and 0 photons in the other half. The frequency corresponding to such a state will be
$$\sum_{i=1}^{N/2}2\cdot 3^{i-1}=3^{N/2-1}-1,$$
and the complexity of the algorithm will be equal to
$$O\left(\frac{N^{2}}{2}\left(3^{N/2-1}-1\right)+\left(3^{N/2-1}-1\right)\log\left(3^{N/2-1}-1\right)\right)=O\left(N^{2}3^{N/2}+N3^{N/2}\right)=O\left(N^{2}3^{N/2}\right).$$

For comparison, the complexity of the Clifford–Clifford algorithm (which is $O(M2^{M}+NM^{2})$) in this case will be equal to $O(N2^{N})$.

2. $M=N^{2}$, the goal is to compute the output state that contains 2N photons in one half of the outputs and 0 photons in the other half. The frequency corresponding to such state will be $\sum_{i=1}^{N/2}2N\cdot {\left(2N+1\right)}^{i-1}={\left(2N+1\right)}^{N/2-1}-1$, and the algorithm complexity will be
$$O\left(N^{2}{\left(2N\right)}^{N/2}+N{\left(2N\right)}^{N/2}\log N\right)=O\left(N^{2}\cdot {\left(2N\right)}^{N/2}\right)$$

Again, the complexity of the Clifford–Clifford algorithm, in this case, will be equal to $O(N^{2}2^{N^{2}})$.

### Weighted Average Complexity

We can measure the weighted average computational complexity of the algorithm described above by computing $\sum_{i}p_{i}C_{i}$, where *i* numbers measurement outcomes (output states) for the bosonic sampler, $p_{i}$ is the probability to observe the *i*th state, $C_{i}$ is the complexity of calculating $p_{i}$ (assuming the second method of choosing $\left\{Q_{i}\right\}$ is used), and the sum is calculated over all possible outcomes.

We have computed this weighted average complexity for the systems with varying *N*. We set M=N, and |k〉=|1,1,…,1〉 as the input state. The interferometer matrices for those systems were randomly generated unitary matrices. Figure 2 shows that the weighted average complexity of the algorithm is significantly lower than the maximum complexity of the algorithm and their ratio decreases as *N* increases.

## 6. The Metropolis–Hastings

For systems with large *N* it might be computationally intractable to calculate the exact probability distribution of output states. The number of the possible output states scales with *N* and *M* as
$$C_{M+N-1}^{N-1}=\frac{(M+N-1)!}{(N-1)!(M+N-1-(N-1))!}=\frac{(M+N-1)!}{(N-1)!M!}.$$

Sampling from a probability distribution from which direct sampling is difficult can be done using the Metropolis–Hastings algorithm, which uses a Markov process. It allows us to generate a Markov chain in which points appear with frequencies that are equal to their probability. In our case, the points will be represented by the output states, i.e., sets of numbers $|l_{1},l_{2},\ldots ,l_{N}\rangle$ such that $\sum_{i=1}^{N}l_{i}=M$.

We will require a transition function that will generate a candidate state from the last state in the chain. When the points are represented by real numbers, a candidate state can be chosen from a Gaussian distribution centered around the last point. In our case, however, the transition function will be more complex.

The transition function (Algorithm 1) must allow the chain to arrive in each of the possible states. It will be convenient to define it in the following way:
**Algorithm 1:** Transition function$h:=\{i:\;l_{i}>0\}$               ▹*h* is a set that contains indices of non-empty outputsK:=|h|                         ▹*K* is the amount of non-empty outputsr:=random({1,…,K})              ▹ We generate a random number 1≤r≤K$l_{h_{r}}:=l_{h_{r}}-1$                  ▹ Decrease the number of photons in $h_{r}$-th output$s:=random(\left\{1,\ldots ,N\right\}\setminus h_{r})$    ▹ We generate a random number 1≤s≤N such that s≠r$l_{s}:=l_{s}+1$                    ▹ Increase the number of photons in *s*-th outputreturn $|l_{1},l_{2},\ldots ,l_{N}\rangle$

For the condition of a detailed balance to hold, we will require a function $p(l_{1},\ldots ,l_{N};l_{1}^{\prime },\ldots ,l_{N}^{\prime })$ which is equal to the probability of $|l_{1}^{\prime },\ldots ,l_{N}^{\prime }\rangle$ being the transition function output when the last state in the chain is $|l_{1},\ldots ,l_{N}\rangle$. It is trivially constructed from the transition function.

Let *u* be the ratio of the exact probabilities of the states $|l_{1}^{\prime },\ldots ,l_{N}^{\prime }\rangle$ and $|l_{1},\ldots ,l_{N}\rangle$. The condition of a detailed balance will hold if the Markov chain will go from the state $|l_{1},\ldots ,l_{N}\rangle$ to the state $|l_{1}^{\prime },\ldots ,l_{N}^{\prime }\rangle$ with the probability
$$\alpha =u\cdot \frac{p(l_{1}^{\prime },\ldots ,l_{N}^{\prime };l_{1},\ldots ,l_{N})}{p(l_{1},\ldots ,l_{N};l_{1}^{\prime },\ldots ,l_{N}^{\prime })}.$$

Given the Markov chain, we can then calculate the approximate probability of a state by dividing the number of times this state occurs in the chain by the total number of steps of the chain.

### Results

To demonstrate that the frequencies with which states appear in the Markov chain converge to the exact probability distribution, we have tested it on the system with N=10,M=10,|k〉=|1,1,…,1〉 and a random unitary 10×10 matrix as the interferometer matrix. To calculate the distance between the exact and the approximate distribution we used cosine similarity:$$S_{C}\left(p,q\right)=\frac{\left(\vec{p}\cdot \vec{q}\right)}{|\vec{p}\left|\cdot \right|\vec{q}|}=\frac{\sum_{i}p_{i}q_{i}}{\sqrt{\left(\sum_{i}p_{i}^{2}\right)}\cdot \sqrt{\left(\sum_{i}q_{i}^{2}\right)}},$$
where *p* and *q* are some probability distributions. Namely, the value of $1-S_{C}\left(p,q\right)$ is 0 when *p* and *q* are equal.

Figure 3 shows that $1-S_{C}\left(p,q\right)$ decreases as the Markov chain makes more steps.

## 7. Conclusions

We have presented a new algorithm for calculating the probabilities of the output states in the boson sampling problem. We have shown the correctness of the algorithm and calculated its computational complexity. This algorithm is simple in implementation as it relies heavily on the Fourier transform, which has numerous well-documented implementations.

The performance of this algorithm is better than the other algorithms in cases where there are many photon collisions. An example we give is an output state where all the photons are spread equally across one-half of the outputs, with the other half of the outputs empty. In this case the algorithm requires $O\left(N^{2}3^{N/2}\right)$ operations, while the Clifford–Clifford algorithm requires $O\left(N2^{N}\right)$ operations.

We have also proposed a method to approximately calculate the probability distribution in the boson sampling problem. It can be used when the system size is too large and calculating the exact probability distribution is intractable. Our results show that this algorithm indeed produces a probability distribution that converges to the exact probability distribution.

We plan to study further the application of the Metropolis–Hastings algorithm to approximate the boson sampling problem. When losses are modeled in the system, the probability distribution of the output states becomes concentrated. For example, when losses are high, the most probable states are those with many lost photons. When the losses are low, the probability is concentrated in the area where no or a few photons are lost. This property makes the Metropolis–Hastings algorithm especially effective in solving this problem.

## Figures and Tables

**Figure 1 entropy-25-00210-f001:**
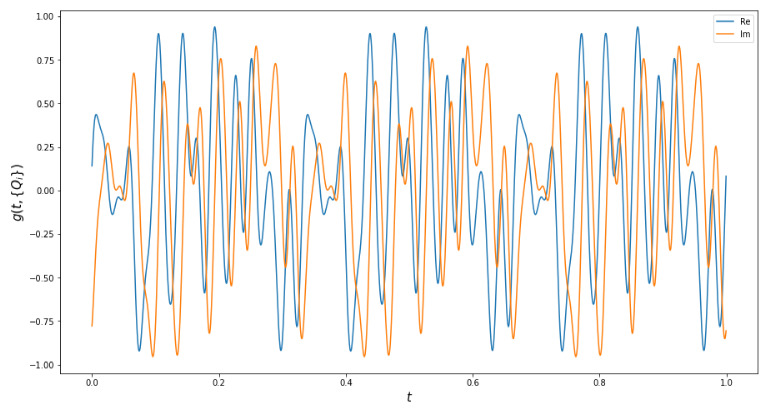
An example of the function $g(t,\left\{Q_{i}\right\})$ (the **top** picture) and its spectrum obtained as squared modulus of the Fourier transform of g(t) (the **bottom** picture). A system with N=M=3 is used, the input state is |k〉=|1,1,1〉. Each peak in the spectrum corresponds to one of ten possible output states.

**Figure 2 entropy-25-00210-f002:**
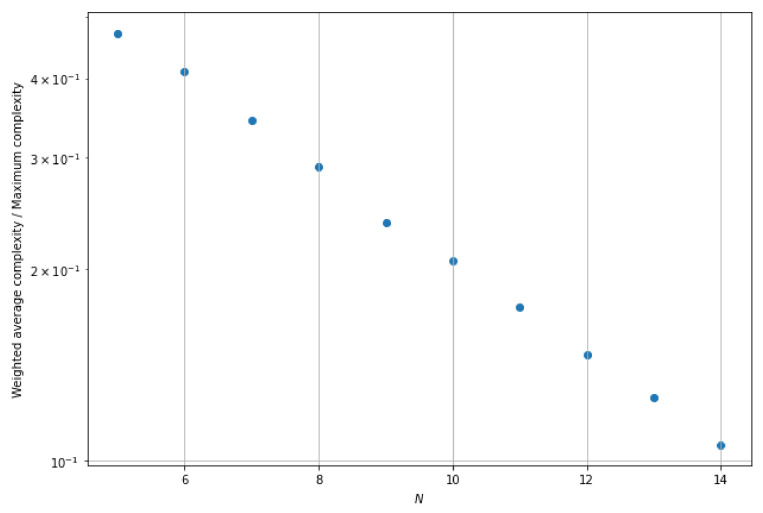
Decrease of the ratio of weighted average complexity to maximum complexity with the increase of *N*.

**Figure 3 entropy-25-00210-f003:**
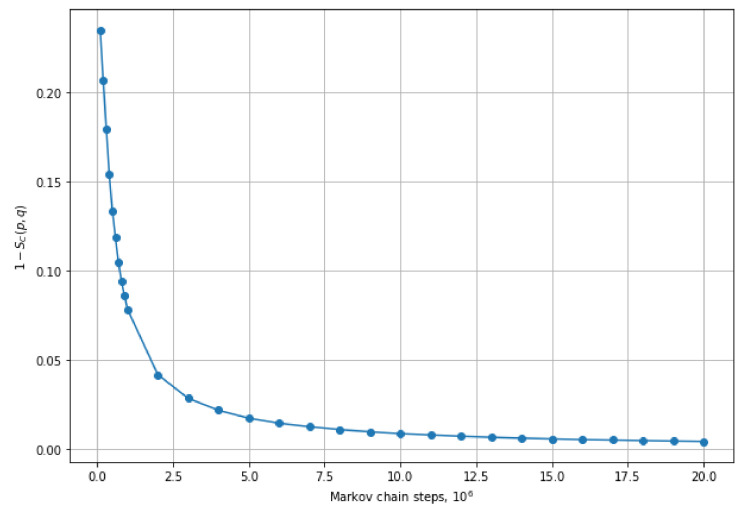
Convergence of the approximate probability distribution to the exact probability distribution.

## Data Availability

Not applicable.

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
