# Peer review of "Classical Modelling of a Bosonic Sampler with Photon Collisions"

_entropy, 2023, doi:10.3390/e25020210_

Round 1
Reviewer 1 Report
I recommend paper for publishing after minor corrections.
Namely, the boson sampling deserves better explanation in the Introduction.
I think that list of References needs to be reconciled with the MDPI format.
Author Response
We are grateful to the Referee for reading our manuscript. Accordingly to the recommendations given by Referee, we extended the Intrduction part to give a better explanation of the boson sampling technique. The format of References was changed as well.
Reviewer 2 Report
The paper addresses an important and complicated question of whether practical/near term boson samplers are available for simulation with classical computers. The paper presents a novel classical algorithm for simulating the probabilities/distributions of sampler outputs. While the complexity of the algorithm is in general exponential, in accordance with known rigorous results, it performs considerably better compared to previous classical algorithms in case of numerous photon collisions. This is a typical case for state-of-the-art samplers.
The paper is written clearly, the results are sound and can be interesting for a broad quantum audience. I believe the paper can be published in its present form.
Author Response
We are grateful to the Referee for reading our manuscript and providing a positive evaluation.
Reviewer 3 Report
Authors presented a new algorithm for calculating the probabilities of the output states in the boson sampling problem (show the correctness of the algorithm and calculated its computational complexity).
Discussed problems are interesting and worth publication.
In text, the spectrum is discussed (also the spectrum is presented in Fig.1). Some explanation how spectrum is calculated wil be valuable.
What probability (line 194) of i-th state is taken into account is not clear.
The paper can be reccomended for publication (optionally after minor revision).
Author Response
We are grateful to the Referee for reading our manuscript. We followed the recommendations: it is now explained how the spectrum is calculated (capture to Fig 1); the probability p_i in chapter 5.1 is defined more clearly.